# Anomalous Photocurrent Reversal Due to Hole Traps in AlGaN-Based Deep-Ultraviolet Light-Emitting Diodes

**DOI:** 10.3390/mi13081233

**Published:** 2022-07-31

**Authors:** Seungyoung Lim, Tae-Soo Kim, Jaesang Kang, Jaesun Kim, Minhyup Song, Hyun Deok Kim, Jung-Hoon Song

**Affiliations:** 1Photonic/Wireless Devices Research Division, Electronics and Telecommunication Research Institute, Daejeon 34129, Korea; sylim93@etri.re.kr (S.L.); sminhyup@etri.re.kr (M.S.); 2School of Electronics Engineering, Kyungpook National University, Daegu 41566, Korea; hyundkim@ee.knu.ac.kr; 3Department of Data Information and Physics, Kongju National University, Gongju 32588, Korea; kts@phovel.com (T.-S.K.); kokonut73@smail.kongju.ac.kr (J.K.); kjsun150910@smail.kongju.ac.kr (J.K.); 4R&D Team, Phovel Co., Ltd., Daejeon 34302, Korea

**Keywords:** AlGaN, deep ultraviolet, light-emitting diode, photocurrent spectroscopy, anomalous photocurrent

## Abstract

The trap states and defects near the active region in deep-ultraviolet (DUV) light-emitting diodes (LED) were investigated through wavelength-dependent photocurrent spectroscopy. We observed anomalous photocurrent reversal and its temporal recovery in AlGaN-based DUV LEDs as the wavelength of illuminating light varied from DUV to visible. The wavelength-dependent photocurrent measurements were performed on 265 nm-emitting DUV LEDs under zero-bias conditions. Sharp near-band-edge (~265 nm) absorption was observed in addition to broad (300–800 nm) visible-range absorption peaks in the photocurrent spectrum, while the current direction of these two peaks were opposite to each other. In addition, the current direction of the photocurrent in the visible wavelength range was reversed when a certain forward bias was applied. This bias-induced current reversal displayed a slow recovery time (~6 h) when the applied forward voltage was removed. Furthermore, the recovery time showed strong temperature dependency and was faster as the sample temperature increased. This result can be consistently explained by the presence of hole traps at the electron-blocking layer and the band bending caused by piezoelectric polarization fields. The activation energy of the defect state was calculated to be 279 meV using the temperature dependency of the recovery time.

## 1. Introduction

Deep-ultraviolet (DUV) light consists of electromagnetic waves with wavelengths of 300 nm or lower and can be used in many applications, such as sterilization, water and air purification, medicine and biochemistry, photolithography, as light sources for high-density optical disks, and for chemical decomposition [1,2,3,4]. The DUV near 265 nm is known to be very effective in sterilizing viruses and has recently attracted more attention due to its effective capacity to break RNA viruses, such as COVID-19. Specific spectral lines from gas discharge lamps, such as xenon and mercury lamps, are currently used in these applications, which is an inconvenient and inefficient process. DUV light-emitting diodes (LEDs) are being used in these applications, as they are compact and highly efficient. Due to their wide direct transition energies in the ultraviolet (UV) range of 3.4 eV (GaN) to 6.2 eV (AlN), AlGaN compounds are being used as the active material in DUV LEDs [5,6,7].

AlGaN-based DUV LEDs still do not have high enough efficiencies for wide application. Their efficiency is lowered as the Al composition is increased to make shorter wavelength UV LEDs. There are several reasons for the decrease in efficiency with a higher Al composition, including high threading dislocation density due to a low lateral growth rate and differences between the lattice constant and the thermal expansion coefficient in the sapphire substrate and the AlN buffer layer. Moreover, p-GaN layers are usually grown on top of p-AlGaN layers since ohmic contact on p-AlGaN is difficult to achieve. A complex electron-blocking layer (EBL) structure is required due to the relatively low p-doping concentration in p-AlGaN compared to in GaN-related blue LEDs. However, the EBL can generate defects around the active region in the UV LEDs [3,4,7,8,9,10,11,12]. Hence, AlGaN DUV LEDs may exhibit many defective states near the active region. This is a major factor impacting the efficiency of LEDs. Studies on the point defects or complex defect states in the active region and their origin have only recently been conducted, and studies on the defects are focused on threading dislocation using electron microscopes.

In this study, the trap states and defects in the active region in AlGaN DUV LEDs are investigated using wavelength-dependent photocurrent spectroscopy. We observed an anomalous photocurrent reversal and its temporal recovery. This phenomenon was attributed to the photocurrent generated by Al vacancies in p-AlGaN and hole traps at the EBL and p-AlGaN interface. Furthermore, we verified our hypothesis by analyzing the change in temporal recovery depending on temperature, and the activation energies of the trap states were obtained.

## 2. Materials and Methods

### 2.1. Sample

We investigated a commercial-grade AlGaN-based 265 nm-emitting DUV LED. Its sample structure is shown in Figure 1. The strains of AlGaN layers grown on a sapphire substrate are strong and have a high dislocation density due to the difference in the lattice constant and the thermal expansion coefficient. Therefore, an AlN buffer layer between the sapphire substrate and the AlGaN layer was included to minimize the strain and dislocation density of the AlGaN to be grown atop.

We grew n-type AlGaN layers on the AlN buffer with an Al composition ratio of 0.7. Multi-quantum wells (MQWs) composed of Al_0.7_Ga_0.3_N barriers and Al_0.53_Ga_0.47_N quantum wells were grown on top of the n-Al_0.7_Ga_0.3_N layer. An electron-blocking layer was placed above the MQWs to block the electron overflow. A p-AlGaN layer was grown on this layer with an Al composition ratio of 0.7, the same as the n-AlGaN layer. The final layer grown was a p-GaN contact layer for p-contact, used as a flip-chip structure to collect the emitted light in the sapphire substrate side.

Figure 2 shows the electroluminescence spectra measured at 10 mA forward current at 300 K and 80 K with a duty cycle of 15 %. At 300 K, the band-edge signal at 265 nm was dominant, and the external quantum efficiency at a maximum output power of 1 mW was measured to be 1%.

### 2.2. Photocurrent Spectroscopy

Photocurrent spectroscopy is a method used to measure photogenerated current depending on the wavelength of excitation light. When a reverse bias is applied to an LED, the carriers generated by light absorption in the MQW layer appear as a reverse current by the built-in electric field in the junction [13,14].

There are two conditions that must be satisfied for absorption to occur in the depletion region or in the MQW region. First, in order for the light to reach the active (depletion) region, the photon energy must be lower than that of the bandgap of the barrier, n-AlGaN, AlN, and sapphire substrates. Second, the photon energy of the incident light must be greater than the bandgap energy of the quantum wells (Al_0.53_Ga_0.47_N). Generally, no photocurrent is observed for the wavelength region longer than the absorption band of the QWs, and a photocurrent does not flow for wavelengths shorter than the absorption wavelength of the barriers (Al_0.7_Ga_0.3_N) since light is absorbed in the thick n-AlGaN layers. Therefore, the expected photocurrent spectrum will exhibit a peak.

The photocurrent spectroscopy setup is shown in Figure 3. As seen in the figure, the measurement setup consists of a xenon lamp, a monochromator, and a source meter. Xenon lamps have an emission range of 200 nm–800 nm. This can be converted to a single wavelength from 200 to 800 nm through a monochromator. A reverse voltage is applied to the sample, and the current is measured through the source meter while irradiating the sample with monochromatic light. The photocurrent spectrum shows the intensity of the photocurrent while changing the wavelength of light from the monochromator. LEDs typically have recovery times of a few µs after illumination [15]. The sweep rate of the monochromator was set to 1 nm in 1 s intervals, and the photocurrent value was obtained 1 s after each wavelength change in order to ensure a static value of the photocurrent.

Additionally, with this setup, it is possible to measure the photocurrent over time while the wavelength of the excitation light is fixed. In order to analyze the time-dependent recovery of the photocurrent reversal shown in the results, we measured the recovery of photocurrent with a time interval of 1 min while we fixed the wavelength of the excitation light at 480 nm when applying back to zero bias after applying a forward bias of 4 V. In addition, in order to confirm the temperature dependence of the photocurrent recovery, the experiments were conducted at 26 °C (room temperature), 50 °C, 100 °C, and 150 °C.

### 2.3. Photoluminescence Measurement

Photoluminescence spectroscopy is a powerful tool to determine the absorption and emission properties of semiconductors. When excitation light greater than the bandgap energy of the absorption band is irradiated, transitions occurring through the defect state as well as the bandgap can be observed. In general, excitation light with an energy greater than the bandgap energy of the part to be observed for PL is used. However, in this paper, a 325 nm He-Cd laser, which is smaller than the bandgap energy of AlGaN QW, was used as an excitation light to find the cause of absorption in the non-QW region. After irradiating the sample with a 325 nm He-Cd laser, we removed the laser signal using a 325 nm notch filter and measured the photoluminescence spectrum from the sample in the range of 300–700 nm.

## 3. Results and Discussion

We measured the photocurrent spectrum from the DUV LED to analyze the absorption in the active region. Figure 4a is the result of measuring the photocurrent spectrum of the DUV LED under zero bias. As mentioned in Section 2.2, the photocurrent spectrum is expected to have a peak in the reverse direction. In Figure 4a, the photocurrent spectrum exhibits a dip between 250 and 270 nm, as expected. However, the photocurrent was also observed at wavelengths above 300 nm, which was unexpected, and the photocurrent was a *forward current* in this case. This result implies that light with wavelengths above 300 nm is also absorbed near the active region in the DUV LED, and the electric field associated with this absorption is opposite to the built-in electric field in the MQW region. This phenomenon has not been observed before in other LEDs, such as InGaAs-based red and InGaN blue LEDs [16,17,18]. In addition, when a forward voltage of 4 V was applied to the DUV LED, as shown in Figure 4b, and returned to the zero bias again, the direction of the photocurrent at wavelengths above 300 nm was reversed, as shown in Figure 4c.

In order to confirm the reversal of the photocurrent observed at wavelengths above 300 nm, the wavelength of excitation light was fixed at 480 nm and a temporal evolution of this photocurrent was measured after the applied bias was returned to the zero bias. The 4 V forward bias was applied for 15 min and then turned off to the zero bias. These results are shown in Figure 5. As seen in the figure, the photocurrent reversed by the 4 V forward bias became smaller and was restored to the positive value after about 6 h.

To summarize the experimental results

A photocurrent flowed at a wavelength above 300 nm at zero bias;A photocurrent flowing at a wavelength above 300 nm is a forward current;When the 4 V forward voltage was applied and then returned to the zero bias, the photocurrent generated at wavelengths above 300 nm was converted into a reverse current;The photocurrent reversal phenomenon was restored over approximately 6 h.

Since a photocurrent occurs at wavelengths above 300 nm, it implies there is a region that absorbs these longer wavelengths. In addition, the absorption band is wide, about 300 to 800 nm. We attribute this wide absorption band to the high point defect densities in these DUV LEDs. If the density of the point defects is high enough, defect states may result in forming the absorption band. 

In order to check the validity of the assumption, the photoluminescence (PL) spectrum of this DUV LED was measured using a 325 nm He-Cd laser as the excitation source, as shown in Figure 6. We note here that the 325 nm line has a lower photon energy than the bandgap energy of the QWs. As seen in Figure 6, a broad strong PL signal was observed with two broad peaks at 400 nm and 545 nm. It was reported that the Al vacancies in the AlN have deep-level defects that can cause emission at 387 nm and 445 nm and absorption at 375–590 nm [19,20,21]. The observed PL peaks had a longer wavelength than the previously reported emission wavelength of Al vacancies in AlN. However, it should be noted that this result was observed in AlGaN-related DUV LEDs with Al composition ratios of 0.53 to 0.7. Since the bandgap energy of AlGaN is smaller than that of the AlN, the deep-level energy of the Al-vacancies may also be small. Our results imply that the defect-related below-gap photoabsorption should occur in the EBL or the EBL/p-AlGaN interface to account for the electric field reversal, which means the related defects should be spatially localized in these very thin layers. Therefore, we believe the photo-absorbing defect band is likely generated by the point defects, not by extended defects such as threading dislocations. In AlGaN, Al vacancy, N vacancy, substitutional oxygen, and the Al vacancy–oxygen complex are known to generate deep-level states [19,20,21]. Considering the central wavelength of the PL peak, it is assumed that Al vacancy or the Al-vacancy–oxygen complex is the dominant cause for these point defects [19,20,21], while further study is needed to specify the exact origin of these defects.

Since a forward current was observed at a wavelength above 300 nm, absorption occurred in a region other than the active region where the built-in electric field generated by the junction is located. This implies that an electric field in the opposite direction to the built-in electric field exists in the absorption layer. Since nitride compound semiconductors have a Wurtzite structure, a strong piezoelectric field (~ 1 MV/cm) exists when the structure deforms due to a difference in the lattice constant between the growth layers [22,23]. The EBL has a high Al content to block electron overflow and, therefore, has a smaller lattice constant than p-AlGaN. p-AlGaN, which has a relatively large lattice constant, can be subjected to compressive strain, and the direction of the formed piezoelectric field is in the opposite direction to the built-in electric field generated at the PN junction. Simulation results show that the piezoelectric field is generated by the strain between the QW and barrier or the EBL and p-AlGaN layers in AlGaN-based DUV LEDs [24]. It was shown that the direction of the piezoelectric field generated by the EBL in the p-AlGaN layer was in the opposite direction to that of the built-in electric field [24]. Therefore, we conclude that the photocurrent above 300 nm is generated in the Al vacancy present in p-AlGaN and flows as a forward current due to the piezoelectric field.

We investigated the temperature dependence of the recovery time to find the origin of the photocurrent’s reversal at wavelengths over 300 nm in more detail. Measurements were made at four different temperatures: 26 °C (room temperature), 50 °C, 100 °C, and 150 °C. After the 4 V forward bias was applied at 15 min, as shown in Figure 5, the photocurrent was measured over time in the excitation light of 480 nm at zero bias. The results are shown in Figure 7. These show that the photocurrent recovery exhibited an exponential-like decay, and the recovery time decreased with increasing temperatures. This photocurrent reversal phenomenon by forward bias disappeared at 150 °C and above. This means that all the trapped holes were thermally activated above 150 °C. The recovery time constant was calculated by fitting the photocurrent recovery to the double exponential decay, as in Equation (1), and the result is shown in Table 1.
(1)$$I\left(t\right)=A_{1}e^{-\frac{t}{\tau_{1}}}+A_{2}e^{-\frac{t}{\tau_{2}}}+I_{0}$$

The double exponential decay implies the existence of two trap levels. As a result, the time constants decreased as the temperature increased and went to 0 when *τ*_1_ and *τ*_2_ equaled 100 °C and 150 °C, respectively. The exponential temperature dependence of the recovery time implies that the cause of the electric field reversal is associated with the charge traps. The trap states can be charged by the leakage current at 4 V of forward voltage, which is smaller than the turn-on voltage of the device. In this device, a significant leakage current was observed at 4 V of forward voltage. Carriers trapped in a deep-level state can create an electric field around themselves. As the trapped charge thermally escapes over time, the electric field strength decreases, and the activation becomes faster with increasing temperature. A deep carrier trap occurred in the path of the leakage current, as shown in Figure 8b, which can form an electric field larger than the piezoelectric field in the p-AlGaN layer. The layer in which the trap existed appears to be the EBL and p-AlGaN interface, and the trapped charges were holes, since, in this case, an electric field generated by the trapped charge was in the opposite direction to the piezoelectric field in p-AlGaN, as shown in Figure 8c. When the system was returned to a zero bias, the trapped holes were activated very slowly and exited the trap, returning to their original state, as shown in Figure 8a.

The relationship between the activation energy of the trapped carriers and the temperature is expressed by the Arrhenius equation as
(2)$$\ln\left(\frac{e_{n}}{T^{2}}\right)=-\frac{\Delta ET}{k}+\ln\left(J\right),$$
where *e_n_* is the recombination rate in the trap level, which is reciprocal to the time constant, and Δ*E* is the energy difference between the trap level and the band-edge state closest to the trap level, namely, the activation energy [25,26,27]. We obtained the activation energy by substituting *τ*_2_, measured at three temperature points, into the Arrhenius equation and calculating the slope of the points. As a result, the activation energy of the trap level corresponding to *τ*_2_ was calculated to be 279 meV.

The electric field caused by the traps generated around the active layer compromised the operational stability of the device. For example, charging at the hole-trap layer at the interface between the EBL and p-AlGaN can lower the threshold voltage of the device by affecting the electric field of the active layer but can also block hole injection from p-AlGaN into the active layer. We report that the direct reversal of the photocurrent occurs in the DUV LED, caused by the immediate absorption region changes with the excitation wavelength, not by the direction of the applied electric field. We note here that the direction reversal of the photocurrent, depending on the excitation wavelength, can be potentially applicable in remote optical switches. 

## 4. Conclusions

To analyze the defect states and their related absorption of the active region in DUV LEDs, we obtained photocurrent spectra as a function of applied bias and the sample temperature. We found several abnormal photocurrent reversal phenomena in the spectra. First, a photocurrent was generated in the 300 nm–800 nm region with a photon energy lower than the bandgap energy of AlGaN in the opposite direction to that of the normal photocurrent by QW band-edge absorption. Second, after applying a forward 4 V bias, the direction of the photocurrent was reversed in the 300 nm–800 nm range, and the direction of the reversed photocurrent recovered to its original direction after several hours at room temperature.

To establish the cause of this phenomenon, PL measurements and temperature-dependent recovery time measurements were performed, and the absorption and emission spectra of AlGaN defects were investigated. We found that the cause of the abnormal photocurrent was carrier generation due to absorption in the 300 nm–800 nm wavelength range by the Al vacancies generated near the p-AlGaN and EBL interface. In addition, it was attributed to the hole traps charged by the leakage current in the forward bias at the EBL and p-AlGaN interface, resulting in charging and generating an electric field that was larger than the existing piezoelectric field. 

The activation energy of the hole trap state was obtained by the temperature-dependent recovery time with the Arrhenius equation calculation. We found the activation energy of the trap level to be 279 meV.

As a consequence of the above, AlGaN-based DUV LEDs have a high defect density that generates an unexpected electric field, which can adversely affect the operating characteristics, such as the efficiency, threshold voltage, and the leakage current of the device. Our results suggest that it is preferably required to minimize the point defect, such as Al vacancies in the EBL layer and/or at the EBL and p-AlGaN, rather than the dislocations or deep defects in the n-AlGaN and buffer layers, in order to avoid this unexpected sheet charge generation in AlGaN-based DUV LEDs. We believe our results, the wavelength-dependent reversal of the photocurrent, can also be utilized for long-range remote optical switches, such as in free-space laser satellite communication, where the contrast and reliability of the on/off signal are important since the on/off signal can be determined by the direction of the photocurrent other than the current amplitude.

## Figures and Tables

**Figure 1 micromachines-13-01233-f001:**
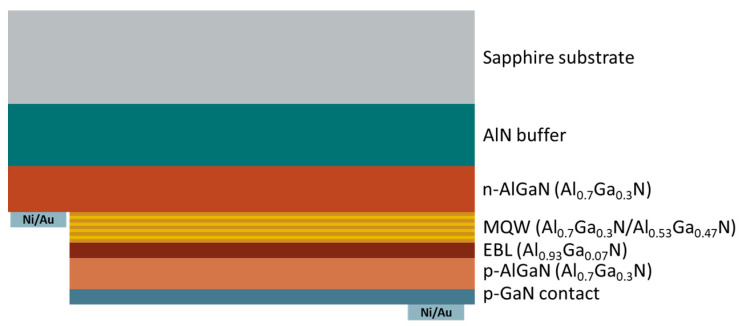
A schematic diagram of the AlGaN-related 265 nm flip-chip deep-ultraviolet (DUV) light-emitting diode (LED) sample.

**Figure 2 micromachines-13-01233-f002:**
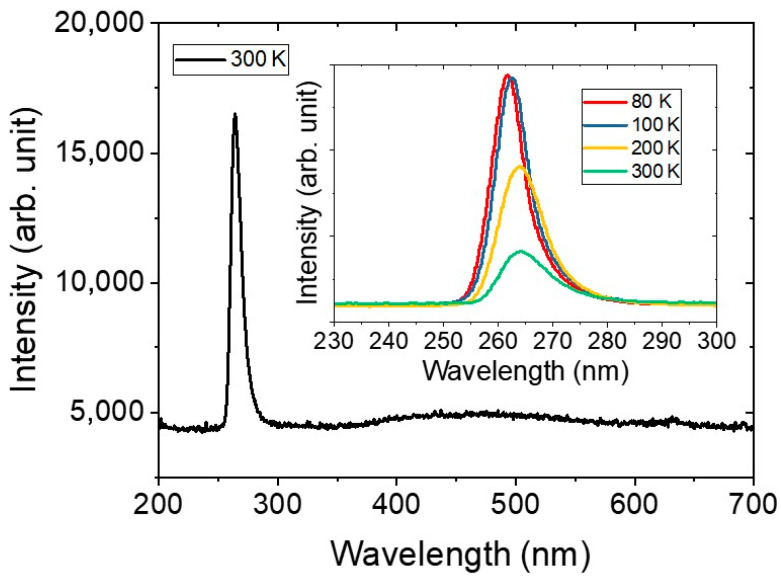
The electroluminescence (EL) spectrum of a 265 nm DUV LED in the range of 200 nm to 700 nm at a temperature of 300 K. The insets are the EL spectra at temperatures of 80 K, 100 K, 200 K, and 300 K.

**Figure 3 micromachines-13-01233-f003:**
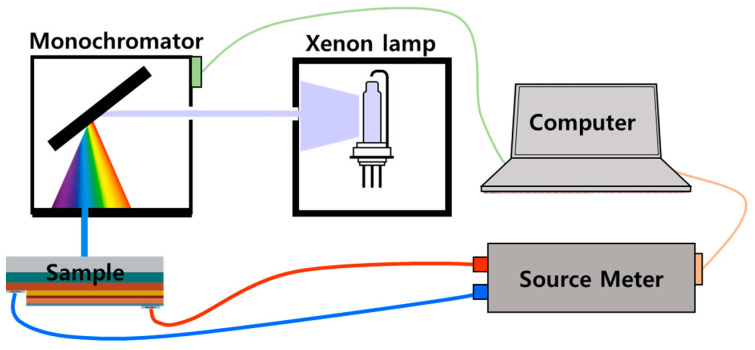
A schematic diagram of the photocurrent spectroscopy measurement.

**Figure 4 micromachines-13-01233-f004:**
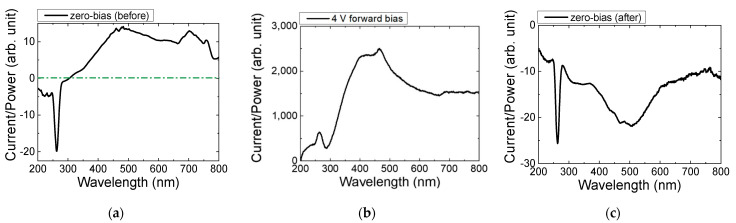
The results of photocurrent spectroscopy divided by lamp power, measured at (**a**) zero bias, (**b**) 4 V forward bias, and (**c**) zero bias remeasured after applying the 4 V forward bias.

**Figure 5 micromachines-13-01233-f005:**
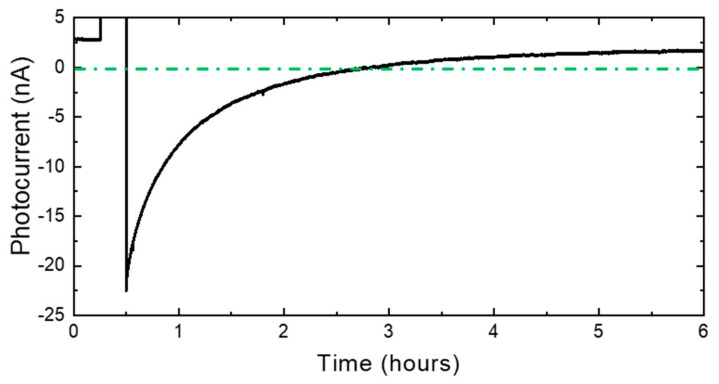
Photocurrent change with time in a 480 nm excitation light. The biases applied were as follows: 0–15 min: zero bias; 15–30 min: 4 V forward bias; 30 min: returned to zero bias.

**Figure 6 micromachines-13-01233-f006:**
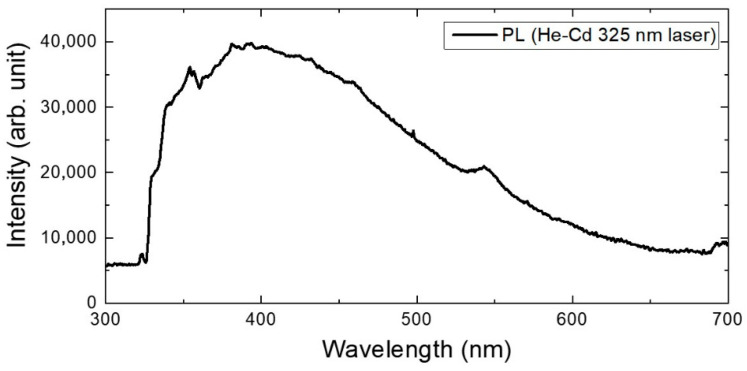
The resulting spectrum of PL spectroscopy using a 325 nm He-Cd laser in a DUV LED.

**Figure 7 micromachines-13-01233-f007:**
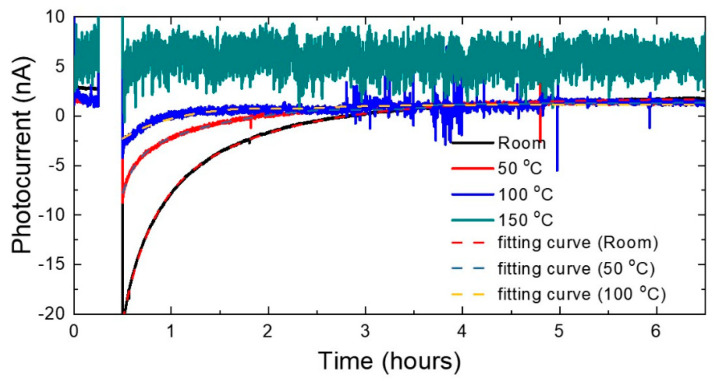
Temperature dependence of the photocurrent change with time. The applied biases are the same as in Figure 5.

**Figure 8 micromachines-13-01233-f008:**
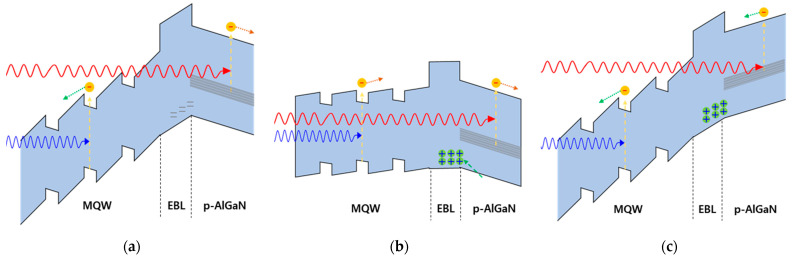
Band diagrams around the active region and the photocurrent mechanism at (**a**) zero bias, (**b**) 4 V forward bias, and (**c**) zero bias remeasured. red arrows: long wavelength above 265 nm; blue arrows: short wavelength less 265 nm.

**Table 1 micromachines-13-01233-t001:** The fitted time constants at each temperature.

Temperature	*τ* _1_	*τ* _2_
26 °C (room temp.)	0.291	1.268
50 °C	0.139	0.999
100 °C	-	0.584

## Data Availability

The datasets used and/or analyzed during the current study are available from the corresponding author on reasonable request.

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
