# Peer review of "Anomalous Photocurrent Reversal Due to Hole Traps in AlGaN-Based Deep-Ultraviolet Light-Emitting Diodes"

_micromachines, 2022, doi:10.3390/mi13081233_

Round 1

Reviewer 1 Report

In this manuscript, the trap states and defects in the active region in AlGaN DUV LEDs are investigated in detail using wavelength dependent photocurrent spectroscopy. Overall, the article is well organized and the results are interesting. I recommend the acceptance of this manuscript for publication in micromachines after the minor revision. I have only one concern. In the conclusion part, the authors said that “AlGaN-based DUV LEDs have a high defect density that generates an unexpected electric field, which can adversely affect the operating characteristics, such as the efficiency, threshold voltage, and the leakage current of the device.” Are there any suggestions or solutions to avoid this unexpected electric field and thus improve the efficiency of AlGaN-based DUV LEDs?

Author Response

첨부 파일을 참조하십시오.

Reviewer 2 Report

In this manuscript, the authors find an anomalous photocurrent reversal phenomenon after applying a forward bias and analyzed the reasons for this photocurrent reversal. The results are interesting, but the practicality or prospects is not well explained in this manuscript. Besides, some conclusions lack solid evidence. Therefore, some issues should be addressed before further consideration for publication.

Below are some concerns:

1.       For photodetectors, it has a recovery time after illumination. Therefore, when you perform the photocurrent spectroscopy measurement, if the wavelength varies very fast, the wavelength-photocurrent result may not be accurate. Please comment on it.

2.       In this manuscript, the authors assume that the photo absorption comes from the Al vacancies in AlGaN layers but lacks solid evidence to support the authors’ opinions (although PL measurement can provide that the light absorption occurs at a longer wavelength, it cannot prove that it comes from the vacancies in AlGaN layers). Please comment.

3.       The abnormal photocurrent reversal is interesting, however, can this phenomenon be used in some special situations? What practical application it will be? The work will be significant if this can be well explained.

4.       The references listed in this article are all out of date and the authors should include recent UVLED papers published in literature, for example: Advanced Functional Materials 29 (48), 1905445, 2019; Optics Letters 46 (13), 3271-3274, 2021 and many others.

Below are some typos that need to be corrected:

1. The descriptions of defect location in DUV LEDs are inconsistent in this manuscript. In the Abstract, “The trap states and defects in the active region in deep ultra-violet (DUV) light emitting diodes (LED) were investigated …”, in the Conclusions, “… absorption in the 300 nm ~ 800 nm wavelength range by the Al vacancies generated near the p-AlGaN and EBL interface”. However, in Figure 7, it is obvious that the trap states and defects are at the interface between the EBL and p-AlGaN.

2. For DUV LEDs, the most important properties are electroluminescence optical power and quantum efficiency. However, there is no relative data in this manuscript so it is difficult to conclude the specific impact of the defects explored in this manuscript on device performance.

3. In the analysis of Figure 6, the authors did not analyze the results under the condition of 150 °C.

4. In part 2.1. Sample, “… an AlN buffer layer between the sapphire substrate and the A1GaN AlGaN layer was included to minimize…”.

5. In part 2.1. Sample, “A multi-quantum wells (MQWs) composed of Al0.7Ga0.3N barriers and Al0.53Ga0.47N quantum wells exists above the AlGaN layer”.

6. In part 2.2. Photocurrent Spectroscopy, “First, the light must reach the active (depletion) region i.e., since absorption must not occur before the depletion region…”.

7. In part 2.2. Photocurrent Spectroscopy, “Second, the photon energy of the incident light must be greater higher than the bandgap energy of the quantum wells…”.

8. On page 4, “We measured the photocurrent spectrum from of the DUV LED to analyze the absorption in the active region”.

9. On page 4, “In order to confirm that the reversal of the photocurrent observed at wavelengths above 300 nm, the wavelength of excitation light was fixed at 480 nm and a temporal evolution of this photocurrent was measured after the applied bias was returned to the zero bias”. The marked part should be a complete clause.

10.           On page 4, “The 4 V forward bias was applied at for 15 min and immediately turned off to the zero bias”.

11.           On page 4, “As is seen in the figure, the reversed photocurrent by the 4V forward bias became…”.

12.           On page 6, “As a result, the time constants decreased as the temperature increased, and went to 0 when τ1 and τ2 equaled 100°C and 150°C, respectively”. This sentence makes no sense.

13.           On page 6, “Carriers trapped in a deep level state can create an electric an electric field around them.”

14.           On page 6, “As the trapped charge activates and escapes over time, the electric field strength decreases, and the activation increases…”.

15.           In the part of Conclusions, “…in the 300 nm ~ 800 nm region with photon energy smaller lower than the bandgap engy of AlGaN…”.

Reviewer 3 Report

The manuscript discusses the point defects in AlGaN-based DUV LEDs by wavelength-dependent photocurrent spectroscopy. The authors observed an unexpected photocurrent around 300800 nm in addition to the sharp near band-edge absorption around 265 nm. They also measured the photoluminescence under below-gap excitation and the temperature dependency of the photocurrent to clarify the cause of the abnormal photocurrent at visible region. They concluded that the Al vacancies generated near the p-AlGaN and EBL interface are the main cause of the abnormal photocurrent. The presented results in the manuscript are quite interesting, and the discussion is fairly well organized. The researchers in the field of AlGaN-based LEDs may be interested in the manuscript. However, the reviewer considers that a serious work is needed to improve the description of each section and the quality of the figures. My comments are listed below.

1. The characteristics of the LEDs used in the manuscript should be described. The AlGaN-based DUV LEDs still do not have high enough efficiency because of the poor quality of the EBL layer as the authors described in the introduction section. The readers cannot understand whether the observed abnormal photocurrent is the characteristic of poor LEDs or not.

2. Please reconsider the structure of the manuscript. The experimental conditions of the PL and the temperature dependence of the recovery time are written in the Results and Discussion section in the manuscript. All experimental setup and conditions should be described in the Materials and Methods section. Please revise the description.

3. Please revise the figures and captions.

(a) The layer thickness should be written in the sample section. The readers cannot understand the description “a simple structure” in the caption of Fig. 1. Please clarify the meaning of “simple.”

(b) The vertical axes in Figs. 3 are ambiguous. The reader cannot understand whether the vertical axis in Fig. 3 (a) can be compared to those of Figs. 3 (b) and (c). The letter “©” in Figure 3 is inappropriate.

(c) Figure 7 should be improved. There is no description of the layers.

4. References list needs to be updated. The most recent work referred in the manuscript was published in 2014. Most relevant and most recent works should be referred. The reviewer considers that the AlGaN-based DUV LEDs have been improved during the last decade. The description “this phenomenon has not been observed before in other LEDs, such as InGaAs-based red and InGaN blue LEDs.” in the line 120 in page 4 needs a proper reference.

5. The authors derived the time constants from the fitting using Eq. (1). The results were summarized in Table 1. However, there appears to be little basis. Please add the description and clarify the validity of the parameters. The reviewer recommends the overlaying of the fitting curves on the experimental results in Fig. 6.

Round 2

Reviewer 3 Report

The authors well revised the manuscript in line with all reviewers.